# Influence of *N*-Methylation and Conformation on Almiramide Anti-Leishmanial Activity

**DOI:** 10.3390/molecules26123606

**Published:** 2021-06-12

**Authors:** Anh Minh Thao Nguyen, Skye Brettell, Noélie Douanne, Claudia Duquette, Audrey Corbeil, Emanuella F. Fajardo, Martin Olivier, Christopher Fernandez-Prada, William D. Lubell

**Affiliations:** 1Départements de chimie, Université de Montréal, C.P. 6128, Succursale Centre-Ville, Montréal, QC H3C 3J7, Canada; anh.minh.thao.nguyen@umontreal.ca (A.M.T.N.); 2196732B@student.gla.ac.uk (S.B.); 2Département de Pathologie et Microbiologie, Université de Montréal, C.P. 6128, Succursale Centre-Ville, Montréal, QC H3C 3J7, Canada; noelie.douanne@umontreal.ca (N.D.); claudia.duquette@umontreal.ca (C.D.); audrey.corbeil@umontreal.ca (A.C.); christopher.fernandez.prada@umontreal.ca (C.F.-P.); 3Department of Microbiology and Immunology, McGill University, Montréal, QC H3A 2B4, Canada; emanuella_fajardo@yahoo.com.br (E.F.F.); martin.olivier@mcgill.ca (M.O.)

**Keywords:** almiramide, leishmaniasis, *N*-methylated peptide

## Abstract

The almiramide *N*-methylated lipopeptides exhibit promising activity against trypanosomatid parasites. A structure–activity relationship study has been performed to examine the influences of *N*-methylation and conformation on activity against various strains of leishmaniasis protozoan and on cytotoxicity. The synthesis and biological analysis of twenty-five analogs demonstrated that derivatives with a single methyl group on either the first or fifth residue amide nitrogen exhibited greater activity than the permethylated peptides and relatively high potency against resistant strains. Replacement of amino amide residues in the peptide, by turn inducing α amino γ lactam (Agl) and *N*-aminoimidazalone (Nai) counterparts, reduced typically anti-parasitic activity; however, peptide amides possessing Agl residues at the second residue retained significant potency in the unmethylated and permethylated series. Systematic study of the effects of methylation and turn geometry on anti-parasitic activity indicated the relevance of an extended conformer about the central residues, and conformational mobility by tertiary amide isomerization and turn geometry at the extremities of the active peptides.

## 1. Introduction

Neglected tropical diseases caused by trypanosomatid protozoan infections, such as human African trypanosomiasis, Chagas disease and leishmaniasis, impact significantly on public health, especially in tropical countries, where their influences are compounded due to poverty, environment, and drug resistance [1,2,3]. Leishmaniasis is caused by an intracellular protozoan belonging to the genus *Leishmania* (*L*.) and transmitted by the phlebotomine sandflies [4]. Upon mammal host infection by way of a sand fly bite, the parasites enter the blood cells in the so-called promastigote stage, then begin to multiply in the amastigote stage spreading to other cells and tissues. Depending on the species, the parasites may disperse via blood and lymph fluid to other body sites, such as the skin and major organs.

With over 1 million new cases occurring annually, around 12 to 15 million people worldwide are infected with leishmaniasis [5,6], which presents as four main forms: cutaneous, mucocutaneous, visceral and post Kala-azar dermal leishmaniasis (CL, MCL, VL and PKDL) [7]. Leishmaniasis causes enlargement of the spleen and liver, anemia [8], skin lesions [9], and destruction of mucous membranes [10]. In addition, controlling co-infection with HIV and visceral leishmaniasis has become a serious challenge [11].

In the absence of a vaccine, chemotherapy is the main tool to treat leishmaniasis, but only a few drugs are used clinically [12]. Pentavalent antimonial drugs, such as sodium stibogluconate and meglumine antimoniate, were first-line defenses against leishmania since the 1940s [13], but are now failing as treatments due to resistance [14]. Amphotericin B, which was discovered in the late 1950s, became an alternative drug for treating leishmaniasis patients [15]. Miltefosine was the first oral anti-leishmanial agent and has been used to treat VL patients since 2002 [16]. High cost and side effects have; however, limited the use of amphotericin B and miltefosine [17]. The aminoglycoside antibiotic paromomycin has been used to treat leishmaniasis [18], with variations in efficacy contingent upon geographic region [19].

The need for new anti-leishmanial agents is critical due to limitations in availability of effective drugs and the rise of drug resistant strains of leishmaniasis [20]. In the search for antiparasitic agents, selectivity and potency (μM) against certain *Leishmania* strains have been observed using chromone natural products [21] and artemisinin-inspired analogs [22]. Moreover, notable differences in ribosomal structure were exploited to conceive paromomycin derivatives with potent anti-leishmanial activity and low ototoxicity [23]. Other anti-leishmanial agents under investigation include mitochondrial cytochrome bc1 and pteridine reductase 1 inhibitors, as well as aryl nicotinic acid derivatives [20]. In addition, the kinetoplastid-selective proteasome inhibitor LXE408 which exhibited efficacy in murine CL and VL models has recently entered Phase 1 human clinical trials [24].

Almiramides A-C (**1**–**3**, Figure 1) are *N*-methylated linear lipopeptides isolated from the marine cyanobacterium *Lyngbya majuscula* [25]. Certain almiramide analogs have exhibited low µM anti-parasitic activity against the causative pathogen of fatal leishmaniasis, as well as a high therapeutic index [25]. Almiramide derivatives are attractive leads for developing agents against leishmaniasis, in part due to their likely mechanism of action involving disruption of the vital energy machinery proteins of the glycosome [26]. A peroxisome-related organelle without mammalian counterpart, the glycosome performs vital metabolic processes in trypanosomatids [27,28]. In search of glycosome targeting agents, the interface of two key transport proteins [peroxin (PEX)14−PEX5] has been blocked by pyrazolo[4,3-c]pyridines which killed effectively the causative trypanosomatids of Human African trypanosomiasis and Chagas disease, *Trypanosoma brucei* and *Trypanosoma cruzi*, and had low mammalian cell cytotoxicity [29,30]. Moreover, an examination of almiramide analogs in affinity capture, and fluorescent microscopy experiments on *T. brucei,* identified the glycosome proteins PEX11 and glycosomal integral membrane protein-5a (GIM5A) as likely targets [26].

Preliminary structure–activity relationship studies on the almiramides have indicated the importance of the unsaturated lipophilic terminus. Almiramides B and C (**2** and **3**, Figure 1) with 2-methyl-7-octynoyl and -7-octenoyl tails were reported to exhibit low μM activity against *Leishmania donovani*, but their 2-methyl-7-oxooctanoyl counterpart almiramide A (**1**) was inactive [25]. Employment of a 6-heptynyl lipophilic terminus and *N*-methy1 valine^3^ in acid **12** and dimethyl amide **13** gave peptides with μM IC_50_ values against *L. donovani*, and improved therapeutic indices with relatively lower cytotoxicity (CC_50_) against mammalian Vero kidney cells: selectivity index (CC_50_/EC_50_) **12** = **13** (50.2) > **2** (21.8) > **3** (17.4) [25]. In this series, the significance of the peptide *C*-terminal was illustrated by the methyl ester counterpart (e.g., **14**) of peptide acid **12**, which exhibited a significant loss of anti-parasitic activity and a relative gain in Vero cell cytotoxicity [25]. In addition, peptide **2** and novel almiramides D-H (**4**–**8**) possessing the common 2-methyl-7-octynoyl *N*-terminus and a distinct *N*-(Me)Ile^4^ residue were isolated together from marine benthic cyanobacteria in a study that found **2** and **9** exhibited toxicity against human gingival fibroblast cells [31].

The active almiramide conformation has been examined by replacement of the Val^3^-Ala^4^ dipeptide moiety with a mannose-derived furanose amino acid in analogs **9**–**11**. Sugar-peptide hybrids **9**–**11** exhibited a similar activity and selectivity index as miltefosine against intra-macrophage amastigotes of *L. donovani* and Vero cells [32]. Conformational analysis of sugar-peptide hybrids **9** and **11,** by a combination of computational and NMR spectroscopic methods, concluded a potential bent structure from a characteristic nuclear Overhauser effect correlation between the phenylalanine *N*-methyl group and the d-proton of the furanose amino acid residue [32].

The intriguing potential of the almiramides for the development of anti-parasitic agents has prompted a more detailed study to ascertain the features responsible for their potency and selectivity. In spite earlier efforts, limited knowledge exists of the relevance of *N*-methylation and conformation for almiramide activity against parasite and host cells. Among the reported almiramide analogs, peptides having only two *N*-methyl residues [e.g., **5**, Val(Me)^1^ and Ile(Me)^4^] to completely permethylated amides (e.g., acid **12**) have been respectively isolated and synthesized. To the best of our knowledge, the activity of unmethylated and singly methylated almiramide peptides has yet to be reported nor has the effectiveness of almiramide analogs against resistant strains of *Leishmania* been examined. A systematic study of almiramide peptides has now been performed to shine light on the importance of tertiary amides and turn conformers for activity against *Leishmania* and resistant strains.

## 2. Results and Discussion

### 2.1. Chemistry

Commencing with the potent and selective lead peptide **12**, a systematic study of the influence of *N*-methyl groups on each of the five amides was begun by the preparation of the corresponding unmethylated peptide acid **15** (Figure 2). Considering that long chain secondary amide derivatives of **12** retained significant anti-parasitic activity and offered potential for the synthesis of conjugates to study mechanism of action [26], the corresponding 1,6-hexanediamine amides **16** and **17** were examined for comparison with permethylated and unmethylated acids **12** and **15**. Subsequently, an *N*-methyl scan was performed on amide **16** to provide singly methylated analogs **18**–**22**. For systematic solid-phase syntheses of *N*-methyl peptides **18**–**22**, the required *N*-methyl amino acids (Fmoc-Ala(Me)-OH, Fmoc-Phe(Me)-OH and Fmoc-Val(Me)-OH) were respectively prepared in solution, according to literature protocols featuring acid-catalyzed condensation of an Fmoc-protected amino acid with paraformaldehyde, to form an oxazolidinone followed by reduction with triethyl silane and trifluoroacetic acid [33].

In the study of biologically active peptides, α-amino γ-lactam residues (Agl residues, Figure 2) so-called Freidinger-Veber lactams have been commonly used to restrict backbone *ω* and *ψ* dihedral angles to favor β-turn conformers, in which the Agl residue situates at the *i* + 1 position [34,35]. The related *N*-amino-imidazol-2-one (Nai) residues offer similar backbone constraint with potential to add substituents at the heterocycle 4- and 5-positions to mimic side chain function with constrained *χ* dihedral angle geometry [35,36]. A biologically active almiramide bent conformer has been proposed based on the activity of sugar-peptide hybrids **9**–**11** [32]. To probe this hypothesis further, Agl and (5-Me)Nai residues were used to replace systematically the first four residues in the sequences of peptides **12** and **15**–**17** (Figure 3). For example, six Agl almiramide derivatives (e.g., **23**–**28**) were respectively prepared by replacing valine (e.g., Val^1^, Val^2^, Val^3^) with the heterocycle in acids **12** and **15**. The first four residues of permethylated and unmethylated peptide amides **16** and **17** were also systematically replaced by Agl residues to provide peptides **29**–**35**. Moreover, (5-Me)Nai residues were respectively used to replace Val^1^ and Val^2^ of peptide acid **15** in peptides **36** and **37**.

For the Agl scan, Fmoc-Agl dipeptides [Fmoc-Agl-Val-OH, Fmoc-Agl-Ala-OH and Fmoc-Agl-Phe-OH (**38a**–**c**), Appendix A] were respectively synthesized by a modification of the original Freidinger-Veber protocol, as recently reported for the preparation of the valine dipeptide [34,37]. In brief, *N*-(Boc)methionyl dipeptide *tert*-butyl esters were treated with iodomethane to prepare the corresponding sulfonium ion intermediate, which on treatment with NaH underwent intramolecular *N*-alkylation to furnish *N*-(Boc)Agl dipeptide *tert*-butyl esters (e.g., **39a**–**c**, Appendix A). The carbamate and ester groups were removed using trifluoroacetic acid (TFA) in dichloromethane and the Fmoc group was installed using *N*-(9-fluorenylmethoxycarbonyloxy)succinimide (Fmoc-OSu) and sodium carbonate in aqueous acetone to provide Agl dipeptides **38**. During the cyclization to form Boc-Agl-Phe-O*t*-Bu (**39c**, Appendix A), ester epimerization occurred providing an inseparable 2:1 mixture of diastereomers, which was used to prepare the separable mixture of Agl-Phe peptide **32** and Agl-D-Phe isomer ***R*-32**. The corresponding Nai-dipeptide ester [Fmoc-(5-Me)Nai-Val-O*t*-Bu (**40**, Appendix A)] was synthesized from Fmoc-azaGly-Val-O*t*-Bu (**41**, Appendix A), by a route featuring oxidation to the corresponding azopeptide using *N*-bromosuccinimide (NBS) and lutidine in dichloromethane, proline-catalyzed alkylation with propionaldehyde, and dehydration using *p*-toluenesulfonic acid in chloroform [38]. Ester solvolysis using TFA in dichloromethane gave the Fmoc-(5-Me)Nai-Val-OH which without purification was coupled onto resin.

As illustrated by the synthesis of Agl analog **24**, peptides **15**–**37**, all were synthesized using conventional Fmoc/*t*-Bu solid phase peptide synthesis (SPPS) methods starting from chlorotrityl resin (CTC resin, Scheme 1) [25,39]. *N*-Methyl peptide amides **18**–**22** were synthesized on CTC resin modified with a 1,6-diaminohexane linker. 1,6-Diaminohexane was loaded onto the CTC resin using *N,N′*-diisopropylethylamine.

*N*-(9-Fluorenylmethoxycarbonyl)phenylalanine was coupled to CTC resin using *N,N’*-diisopropylethylamine. Alternatively, Fmoc-Phe-OH and Fmoc-Phe(Me)-OH were coupled to 1,6-diaminohexane CTC resin using *N,N’*-diisopropylcarbodiimide (DIC) and 1-hydroxybenzotriazole (HOBt) in NMP. After Fmoc group removals with a 20% solution of piperidine in DMF, Fmoc-Ala-OH, Fmoc-Val-OH, Fmoc-Agl dipeptide acids 38a-c, Fmoc-(5-Me)Nai-Val-OH and 6-heptynoic acid, all were respectively coupled using a similar DIC/HOBt protocol. The *N*-methyl residues (e.g., Fmoc-Ala(Me)-OH and Fmoc-Val(Me)-OH) were coupled using 1-[bis(dimethylamino)methylene]-1H-1,2,3-triazolo[4,5-b]pyridinium 3-oxide hexafluorophosphate (HATU) and *N*-methylmorpholine in DMF [40]. Peptide cleavage was performed using a TFA/TES/H_2_O (95:2.5:2.5) cocktail. Final peptides were precipitated with diethyl ether (Et_2_O) and purified by HPLC on a C_18_ column (Table 1). Peptide acids (e.g. **24**) were permethylated using sodium hydride and iodomethane in THF (Scheme 2) [25]. As illustrated for the synthesis of amide **34**, unmethylated and permethylated peptide amides **16** and **17**, as well as Agl peptide amides **29**-**35**, all were prepared by coupling the corresponding peptide acids (e.g., **27**) to 1,6-diaminohexane using *N,N′*-dicyclohexylcarbodiimide and 4-dimethylaminopyridine in dichloromethane.

### 2.2. Bioactivity

The biological activity of peptides **15**–**37** was assessed against the *L**. infantum* wild-type stain (WT), as well as mutants resistant to the common anti-leishmanial agents, such as antimony (Sb^III^), amphotericin B (AmB) and miltefosine (MF): Sb2000.1, AmB1000.1 and MF200.5. Activity against *Leishmania* promastigotes was determined by monitoring the replication of parasites after 72 h of incubation at 25 °C in the presence of increasing concentrations of the different peptides **12**–**37** and reported as EC_50_ values (Table 2). In general, almiramide peptides **12**–**37** exhibited activity against wild type and resistant strains in the range of 5–300 μM. The greatest potencies were observed using analogs against the amphotericin B resistant strain (e.g., AmB1000.1).

Examining peptide potency (EC_50_) against wild type *L. infantum*, the 1,6-diaminohexane amides (e.g., **16** and **17**) were respectively more potent than the acid counterparts (e.g., **15** and **12**). In the Agl series, amides **29**–**31** and **33**–**35** were similarly more active than their acid counterparts **23**–**25** and **26**–**28**. Moreover, the unmethylated analogs (e.g., **15**, **16**, **29** and **31**) were typically more potent than the respective permethylated counterparts (e.g., **12**, **17**, **33** and **35**), except in the case of Agl^2^ analogs **30** and **34**. Comparison of the activity of unmethylated 1,6-diaminohexane amide **16** with singly methylated counterparts **18**–**22** illustrated decreasing potency in the order of **16** > MePhe^5^ **22** > MeVal^1^
**18** > MeVal^3^ **20** > MeAla^4^ **21** > MeVal^2^
**19**. Considering the Agl analogs as constrained versions of the corresponding *N*-methyl dipeptide amides, except in the case of MePhe **22** which was significantly more potent than Agl-Phe **32**, the more restricted Agl analogs **29**–**31** were slightly more active than their *N*-methyl counterparts **19**–**21**. Activity against *Leishmania* dropped significantly in the case of the Agl peptide acids (**23**–**28**) and their (5-Me)Nai counterparts (**36** and **37**).

Many of the structure–activity relationships, that were observed in the wild-type strain, were also found in the resistant strains. For example, the amides were more potent than their acid counterparts. The nonmethylated analogs were also typically more potent than their permethylated analogs; however, in the case of the amphotericin B resistant strain (e.g., AmB1000.1), permethylated analogs **17** and **35** were respectively 4- and 1.4-fold more active than nonmethylated counterparts **16** and **31**, and in the case of the miltefosine resistant strain (e.g., MF200.5), **35** was 1.5-fold more active than **31**. Although the order of potency for nonmethylated **16**, MeVal^1^
**18** and MePhe^5^ **22** was contingent on the tested resistant strain, analogs with no tertiary amide and methylation at the *C*- and *N*-terminal residues were consistently more active than peptides **19**–**21** with *N*-methylation in the central residues. The potency of MePhe^5^ **22** was greater than Agl-Phe^5^ **32** for all strains tested. The Agl constraint at other positions tended to slightly favor activity compared to the tertiary amide counterpart in the antimony resistant strain (e.g., Sb2000.1) as was observed in wild type; however, the lactam analogs were significantly less active than the respective *N*-methyl counterparts against the miltefosine and amphotericin B strains. In addition, preliminary investigations of peptides (e.g., **12**–**22**), which had notable potency against Leishmania (EC_50_ = 5–90 μM), detected no activity against *T*. *brucei* and *T*. *cruzi* (data not shown).

The host cytotoxicity was evaluated by monitoring the survival rates of murine LM-1 macrophage at different concentrations of peptides **15**–**37** (0.0001 to 100 mM) over 24 h. Most analogs exhibited CC_50_ values between 170–450 μM. Exceptionally, MeAla peptide amide **21** was 3.2- to 8-fold less toxic than the other peptide analogs. Comparing macrophage toxicity with potency against *L. infantum*, the selectivity index (SI) for MeAla peptide amide **21** was relatively high (SI = 19.1) and comparable with the most potent peptide (e.g., **16**, SI = 19.2). In comparisons of macrophage toxicity with potency against the AmB resistant *Leishmania* strain, permethylated and MeVal^1^ peptide amides **17** and **18** exhibited, respectively, therapeutic indices of 64 and 36.

## 3. Discussion

The structure–activity relationship studies obtained in wild type and resistant strains of *L. infantum* reflect likely a combination of ability to engage the target and pharmacokinetic properties that may influence peptide availability. The molecular targets of almiramide C have been studied using a combination of photo-affinity and fluorescent probes in *T. brucei*, and suggested to include integral membrane proteins found in glycosomes (e.g., GIM5 and PEX11), which are specific to kinetoplastid parasites [26]. Responsible for the first seven steps of glycolysis, the glycosome is a peroxisome-related organelle essential for parasite survival in the bloodstream stage [41]. Notably, translocation across the glycosomal membrane implicates transporter and pore-forming proteins [42], which may differ contingent on species and be modified in resistant strains.

Resistant strains of *Leishmania* emerge by different mechanisms which contingent on the drug act commonly to reduce the active concentration inside the parasite by either decreasing uptake, increasing efflux or inhibiting activity [43]. Leishmania antimonial resistance is associated with thiol metabolism to prevent reduction of the Sb^V^ to more active Sb^III^, and to sequester antimony in thiolate complexes amenable for efflux [44]. Miltefosine resistance is commonly associated with mutations in the *Leishmania* miltefosine transporter (LMT), a P-type ATPase responsible for the translocation of phospholipids, as well as overexpression of ABC transporters [44]. Amphotericin B resistant *L. donovani* promastigotes have been shown to feature substitution of ergosterol for another sterol, which alters the fluidity and AmB binding affinity in the cell membrane [45].

Although the conformational preferences of the almiramides alone and target bound have yet to be described, information gleaned from related peptides and their *N*-methyl and lactam counterparts offers a lens through which to interpret the structure–activity relationships. For example, the circular dichroism spectrum of the (Val-Val-Val-Ala)_n_ oligomer in a mixture of hexafluoro-2-propanol and trifluoro ethanol indicated a curve shape typical of an extended β-sheet structure [46]. The corresponding Val-Val-Val-Ala region in nonmethylated almiramide analogs **15** and **16**, as well as MePhe analog **22** may likely adopt an extended β-strand conformer. Introduction of *N*-methyl groups causes significant consequences on peptide conformation, due in part to creation of a tertiary amide which loses a potential NH hydrogen-bond donor and may adopt energetically similar *cis* and *trans* isomers [47]. Computational analysis of the *N*’-methyl amides of *N*-acetyl*-N*-methyl- and *N*-acetyl-alanine indicated that repulsive interactions between the *N*-methyl and carbonyl oxygen moieties of the former abolished the low-energy minimum β-conformer adopted by the latter [48]. In cyclic peptides, *N*-methyl residues have also induced backbone *f* and *ψ* dihedral angles consistent with β- and γ-turn conformers [49], as well as altered side chain *χ* geometry [50]. *N*-Alkylation of the central amide of the hairpin inducing D-Pro-Aib dipeptide has also been shown by variable temperature CD spectroscopy to reinforce the central turn conformer and enhance the stability of the folded β-sheet peptide [51]. Introduction of *N*-methyl residues within the peptide chain causes likely a shift from an extended sequence to a dynamic series of *cis*- and *trans*-amide conformers exhibiting a preference for turn geometry, which in peptides **19**–**21** reduces activity. In MeVal^1^ analog **18** and permethylated analogs **12** and **17**, methylation at the *N*-terminal enables a *cis*-amide conformer in which the lipid tail may fold in the direction of the peptide. Such a geometry may improve membrane transport by hiding hydrophilic NH moieties and may account for the significantly improved activity of **17** and **18** against the amphotericin B resistant strains. To further examine the influence of *N*-methylation on conformation, the ^1^H NMR spectra of nonmethylated peptide **16** and *N*-methyl counterpart **18** were examined in DMSO-*d*_6_ (Appendix A). In contrast to peptide **16**, which exhibited a narrow distribution of amide protons signals between 7.55 and 7.95 ppm characteristic of a linear conformer, the corresponding peaks were downfield shifted, dispersed between 7.65 and 8.35 ppm, and existed in isomeric pairs for MeVal^1^ peptide **18**, likely due the tertiary amides favoring *cis*- and *trans*-amide isomers of similar energy and local turn conformers with intramolecular hydrogen bonds.

The propensity for Agl bearing peptides to adopt type II and II’ β-turns contingent on α-carbon stereochemistry has been demonstrated using spectroscopic and computational methods, as well as X-ray diffraction, which has also characterized crystals of lactam analog in extended conformers [52]. Relatively diminished activities of Agl analogs **29**–**35** compared to nonmethylated peptide **16** may again be due to the favored turn geometry. The slightly better activity exhibited by Agl analogs **29**–**31** in comparison to *N*-methyl counterparts **19**–**21** in wild type *L. infantum* may be attributed to the capacity of the amino lactam to stabilize *trans*-amide isomers. On the other hand, the notably better activity of MePhe analog **22** relative to Agl counterpart **32** indicates the attributes of greater conformational flexibility at the *C*-terminal. Similarly, greater conformational dynamics of *N*-methyl analogs compared to the Agl counterparts appear to be important for the relatively better activity of the former in resistant *Leishmania* strains.

## 4. Materials and Methods

### 4.1. Experimental Section

#### 4.1.1. Leishmania Cultures and Antileishmanial Activity Determination

The *L. infantum* (MHOM/MA/67/ITMAP-263) wild-type stain (WT), as well as the resistant mutants Sb2000.1, AmB1000.1 and MF200.5 [53,54,55,56,57,58], resistant to antimony (Sb^III^), amphotericin B (AmB) and miltefosine (MF), respectively, were grown in M199 medium at 25 °C supplemented with 10% fetal bovine serum, 5 μg/mL of haemin at pH 7.0 and 2000 μM Sb (Potassium antimonyl tartrate, Sigma-Aldrich), 200 μM of MF (Miltefosine, Cayman Chem., Ann Arbor, MI, USA) or 1 μM AmB (Amphotericin B solution, Sigma, Oakville, ON, Canada). Antileishmanial values were determined in *Leishmania* promastigotes by monitoring the replication of parasites after 72 h of incubation at 25 °C in the presence of increasing concentrations of the different peptides by measuring A_600_ using a Cytation 5 machine (BioTek, Winooski, VT, USA). EC_50_ values were calculated based on dose–response curves analyzed by non-linear regression with GraphPad Prism 8.4.3 software (GraphPad Software, La Jolla, CA, USA). An average of at least three independent biological replicates from independent cultures was performed for each determination.

#### 4.1.2. LM-1 Macrophages and Cytotoxicity Determination

The LM1- macrophages were grown in DMEM supplemented with 10% heat inactivated FBS. Cells at the concentration of 100,000 cells/mL were cultivated for 24 h in a 96-well plate (37 °C, 5% CO_2_). The culture medium was removed and fresh medium containing the appropriate drug and concentration was added to the cells, which were incubated for 24 h (37 °C, 5% CO_2_). Seven different concentrations were tested (100, 10, 1, 0.1, 0.01, 0.001 and 0.0001 mM) as well as controls (without drugs). After 24 h, the culture medium was removed and replaced for fresh medium containing 10% Alamar Blue (Thermo Fisher Scientific, Waltham, MA, USA) and incubated for 2 h (37 °C, 5% CO_2_). Readings at 570 and 600 nm (Asys UVM340 Plate reader, Biochrom, Cambridge, UK) were taken and analyzed according to the manufacturer protocol. Survival rates at different drug concentrations and CC_50_ values were calculated using the Excel software.

### 4.2. Materials

Anhydrous solvents (THF, DMF, CH_2_Cl_2_, and NMP) were obtained by passage through solvent filtration systems (GlassContour, Irvine, CA, USA). Unless specified otherwise, all reagents from commercial sources were used as received. 2-Chlorotrityl chloride (CTC) resin (1.46 mmol/g, 100–200 mesh), *N*-(9-fluorenylmethoxycarbonyloxy)succinimide (Fmoc-Osu), *N*,*N*′-diisopropylcarbodiimide (DIC), all were purchased from ChemImpex; 4-dimethylaminopyridine (DMAP), *N,N*′-dicyclohexylcarbodiimide (DCC), iodomethane, *N,N*-diisopropylethylamine (DIEA), formic acid (FA), all were purchased from Aldrich; amino acids, such as Fmoc-Phe-OH, Fmoc-Ala-OH, Fmoc-Val-OH, and 6-heptynoic acid were purchased from GL Biochem, ChemImpex and Combi-blocks; solvents were obtained from Fisher. Sodium hydride (60% dispersion in mineral oil) was washed with hexane three times to remove oil prior to use. The *N*-methyl amino acids, Fmoc-Phe(Me)-OH, Fmoc-Ala(Me)-OH and FmocVal(Me)-OH, were prepared by according to the literature procedure and exhibited ^1^H NMR spectra data identical to that previously reported [33]. The Agl dipeptide, Fmoc-Agl-Val-OH was prepared according to the literature procedure and exhibited a ^1^H NMR spectrum identical to that previously reported [37].

Chromatography was on 230−400 mesh silica gel. Analytical thin-layer chromatography (TLC) was performed on glass-backed silica gel plates (Merck 60 F254). Visualization of the developed chromatogram was performed by UV absorbance or staining with ninhydrin. ^1^H and ^13^C NMR spectra were measured respectively in CDCl_3_ at 500 MHz and 126 MHz, and referenced to CDCl_3_ (7.26 and 77.0 ppm). Coupling constants, *J* values were measured in Hertz (Hz) and chemical shift values in parts per million (ppm). Specific rotations, [α]_D_ were measured at 25 °C at the specified concentrations (*c* in g/100 mL) using a 1 dm cell on a PerkinElmer Polarimeter 589 and expressed using the general formula: [α_D_^25^] = (100 × α)/(d × c). High resolution mass spectral analyses were obtained by the Centre Régional de Spectrométrie de Masse de l’Université de Montréal. Protonated molecular ions [M + H]^+^ and sodium adducts [M + Na]^+^ were used for empirical formula confirmation.

Almiramide peptide analog synthesis was performed using Fmoc-based solid-phase synthesis in an automated shaker commencing with 2-chlorotrityl chloride resin. All final peptides were purified on a preparative column (C18 Gemini column) using a gradient from pure water (0.1% FA) to mixtures with MeOH (0.1% FA) at a flow rate of 10 mL/min. Purity of peptides (>95%) was evaluated using analytical LC−MS on a 5 μM 50 mm × 4.6 mm C_18_ Phenomenex Gemini column in two different solvent systems: water (0.1% FA) with CH_3_CN (0.1% FA) and water (0.1% FA) with MeOH (0.1% FA) at a flow rate of 0.5 mL/min using the appropriate linear gradient.



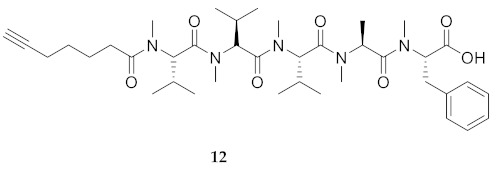



**HCC(CH_2_)_4_CO-Val(Me)-Val(Me)-Val(Me)-Ala(Me)-Phe(Me)-OH (12)** Under argon, peptide **15** (200 mg) was dissolved in THF (40 mL), cooled to 0 °C, treated with NaH (600 mg, 80 eq.), stirred for 5 min, and treated dropwise with iodomethane (0.6 mL, 9 mmol, 30 eq.). The cooling bath was removed. The reaction mixture warmed to room temperature. After 2 h, more iodomethane (0.6 mL, 30 eq.) was added to the reaction mixture, which was stirred for 20 h. The suspension was quenched with water, concentrated to a reduced volume, and acidified to pH = 1 using 10% aqueous HCl. The acidified mixture was extracted 3 times with EtOAc. The organic layers were combined, dried over Na_2_SO_4_, filtered and evaporated to 174 mg of residue, from which part (87 mg) was used to make peptide **17** as described below, and the remainder was purified by HPLC on a C_18_ column using a gradient of 30% to 90% MeOH in H_2_O to obtain peptide **12** (16 mg, 13%), which was shown to be >99% pure by LC-MS analysis [30−95% MeOH (0.1% FA) in H_2_O (0.1% FA), 14 min, RT 6.83 min].



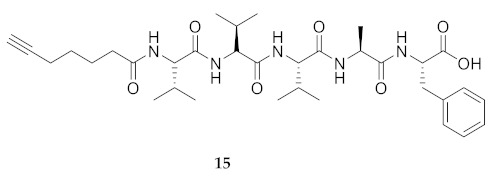



**HCC(CH_2_)_4_CO-Val-Val-Val-Ala-Phe-OH (15)** Fmoc-Phe-OH (848 mg, 1.5 eq.) and DIEA (0.8 mL, 3 eq.) were added to a suspension of 2-chlorotrityl chloride resin (1 g, 200–400 mesh, 1% DBV) swollen in CH_2_Cl_2_ (25 mL). The mixture was shaken for 18 h, filtered and washed sequentially with CH_2_Cl_2_ (3 times for 1 min/wash) and DMF (3 times for 1 min/wash). The Fmoc group was removed upon treatment twice for 20 min with a 20% solution of piperidine in DMF (20 mL/g resin). Subsequently, Fmoc-Ala-OH (3 eq.) was coupled to the resin swollen in NMP (25 mL/g resin) using DIC (3 eq.) and HOBt (3 eq.). After shaking for 16 h, the coupling mixture was filtered, and the resin was washed as described above. Subsequent Fmoc group removal and coupling of Fmoc protected amino acid residues were performed using the above protocols. Complete coupling reactions were confirmed by LC–MS monitoring. After coupling of the last residue, the 6-heptynoic acid (3 eq.), DIC (3 eq.) and HOBt (3 eq.) were added to the resin. The mixture was shaken for 18 h, filtered and washed as previously described. Then, the linear peptide was cleaved from the solid support using a solution of TFA/TES/H_2_O (95/2.5/2.5). The volatiles were evaporated. The reduced volume was treated with diethyl ether. The precipitate was collected by centrifugation (1200 rpm), washed with ether and recollected by centrifugation (3 × 10 min). Removal of diethyl ether afforded a colorless solid [2], which gave 387 mg of residue, from which 200 mg was used to make peptide **12** as described below, and 60 mg purified by HPLC on C_18_ column using a gradient of 30% to 90% MeOH in H_2_O to obtain peptide **15** (24 mg, 12%), which was shown to be >99% pure by LC-MS analysis [30−95% MeOH (0.1% FA) in H_2_O (0.1% FA), 14 min, RT 8.33 min].



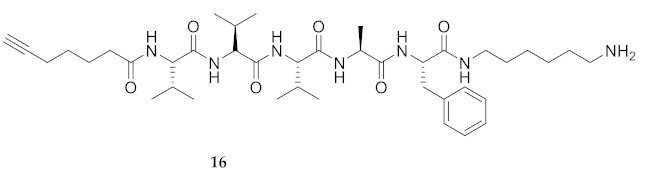



**HCC(CH_2_)_4_CO-Val-Val-Val-Ala-Phe-NH(CH_2_)_6_NH_2_ (16)** A suspension of 2-chlorotrityl chloride resin (250 mg, 200–400 mesh, 1% DBV) swollen in CH_2_Cl_2_ (25 mL) was treated with 1,6-diaminohexane (51 mg, 1.2 eq.) and DIEA (0.2 mL, 3 eq.), shaken for 24 h, filtered and washed sequentially with CH_2_Cl_2_ (3 times for 1 min/wash) and DMF (3 times for 1 min/wash). Subsequently, Fmoc-Phe-OH (3 eq.) was coupled to the resin swollen in NMP (25 mL/g resin) using DIC (3 eq.) and HOBt (3 eq.). After shaking for 16 h, the coupling mixture was filtered, and the resin was washed as described above. Subsequent Fmoc group removals, couplings of Fmoc protected amino acid residues, and acylation with 6-heptynoic acid, all were performed using the protocols described for the synthesis of peptide **15**. The linear peptide was cleaved from the solid support using a solution of TFA/TES/H_2_O (95/2.5/2.5). Removal of the volatiles gave a colorless oil, which was purified by HPLC on a C_18_ column using a gradient of 30% to 90% MeOH in H_2_O to obtain peptide **16** as a white solid (8.8 mg, 3%), which was shown to be >99% pure by LC-MS analysis [30−95% MeOH (0.1% FA) in H_2_O (0.1% FA), 14 min, RT 7.33 min].



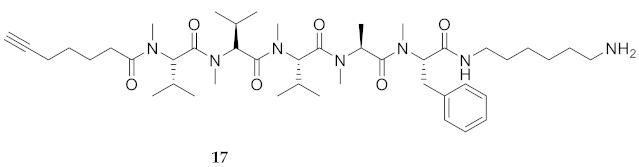



**HCC(CH_2_)_4_CO-Val(Me)-Val(Me)-Val(Me)-Ala(Me)-Phe(Me)-NH(CH_2_)_6_NH_2_ (17)** A solution of permethylated peptide **12** (87 mg) in dry CH_2_Cl_2_ (15 mL) was treated with 1,6-hexamethylenediamine (17 mg, 1.2 eq.), DMAP (4 mg, 0.3 eq.), and DCC (30 mg, 1.2 eq.), and stirred for 48 h at room temperature. The reaction mixture was filtered through Celite™ and washed with CH_2_Cl_2_. The filtrate and washings were evaporated under reduced pressure. The residue was purified by HPLC on a C_18_ column using a gradient of 30% to 90% MeOH in H_2_O. Evaporation of the collected fractions afforded amino hexanamide **17** as white solid (17 mg, 14%), which was shown to be >99% pure by LC-MS analysis [30−95% MeOH (0.1% FA) in H_2_O (0.1% FA), 14 min, RT 9.01 min].



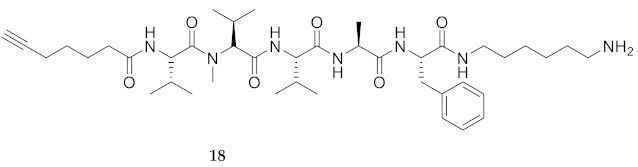



**HCC(CH_2_)_4_CO-Val(Me)-Val-Val-Ala-Phe-NH(CH_2_)_6_NH_2_ (18)** As described for the synthesis of peptide **16**, 1,6-diaminohexane (51 mg, 1.2 eq.) and DIEA (0.2 mL, 3 eq.) were reacted with 2-chlorotrityl chloride resin (250 mg, 200–400 mesh, 1% DBV) in CH_2_Cl_2_ (25 mL). The resulting amine resin was swollen in NMP (25 mL/g resin), treated with Fmoc-Phe-OH (3 eq.), DIC (3 eq.) and HOBt (3 eq.), shaken for 16 h, filtered, and washed as described above. The Fmoc group was removed upon treatment twice for 20 min with a 20% solution of piperidine in DMF (20 mL/g resin). After Fmoc group removal, the peptide was elongated, coupled to 6-heptynoic acid using HATU (3 eq.) in 0.4 NMM in DMF (25 mL/g resin), cleaved and purified as described for the synthesis of peptide **16**. Evaporation of the collected fractions gave peptide **18** (8.2 mg, 3%), which was prepared and shown to be >99% pure by LC-MS analysis [30−95% MeOH (0.1% FA) in H_2_O (0.1% FA), 14 min, RT 8.42 min].



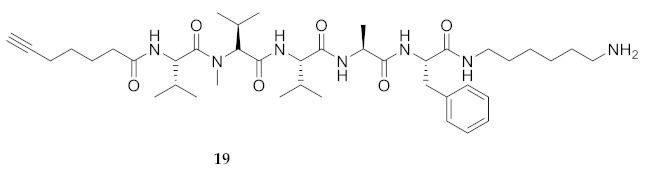



**HCC(CH_2_)_4_CO-Val-Val(Me)-Val-Ala-Phe-NH(CH_2_)_6_NH_2_ (19)** Employing the protocol described above for the synthesis of peptide **18** using Fmoc-Val(Me)-OH (386 mg, 3 eq.) in the fourth coupling, peptide **21** (6.4 mg, 2%) was prepared and shown to be >99% pure by LC-MS analysis [30−95% MeOH (0.1% FA) in H_2_O (0.1% FA), 14 min, RT 7.86 min].



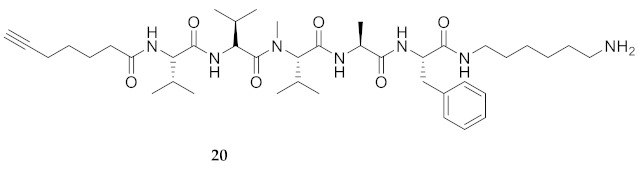



**HCC(CH_2_)_4_CO-Val-Val-Val(Me)-Ala-Phe-NH(CH_2_)_6_NH_2_ (20)** Employing the protocol described above for the synthesis of peptide **18** using Fmoc-Val(Me)-OH (386 mg, 3 eq.) in the third coupling, peptide **20** (5.8 mg, 2%) was prepared and shown to be >99% pure by LC-MS analysis [30−95% MeOH (0.1% FA) in H_2_O (0.1% FA), 14 min, RT 7.91 min].



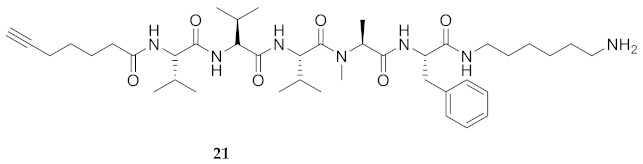



**HCC(CH_2_)_4_CO-Val-Val-Val-Ala(Me)-Phe-NH(CH_2_)_6_NH_2_ (21)** Employing the protocol described above for the synthesis of peptide **18** using Fmoc-Ala(Me)-OH (355 mg, 3 eq.), peptide **21** (7.8 mg, 3%) was prepared and shown to be >99% pure by LC-MS analysis [30−95% MeOH (0.1% FA) in H_2_O (0.1% FA), 14 min, RT 7.61 min]



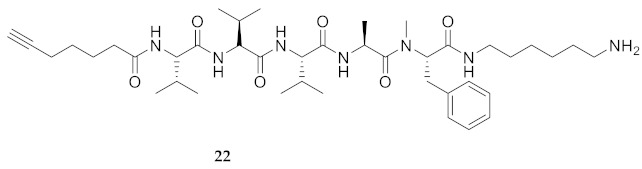



**HCC(CH_2_)_4_CO-Val-Val-Val-Ala-Phe(Me)-NH(CH_2_)_6_NH_2_ (22)** Employing the protocol described above for the synthesis of peptide **22** using Fmoc-Phe(Me)-OH (528 mg, 3 eq.), peptide **22** (4.5 mg, 2%) was prepared and shown to be >99% pure by LC-MS analysis [30−95% MeOH (0.1% FA) in H_2_O (0.1% FA), 14 min, RT 7.62 min].



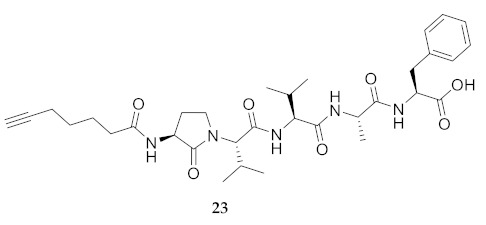



**HCC(CH_2_)_4_CO-Agl-Val-Val-Ala-Phe-OH (23)** Employing the protocol described above for the synthesis of peptide **15** using Fmoc-Agl-Val-OH (462 mg, 3 eq.) in the third coupling, peptide **23** (7.5 mg, 3%) was prepared and shown to be >99% pure by LC-MS analysis [50−95% MeOH (0.1% FA) in H_2_O (0.1% FA), 14 min, RT 7.11 min].



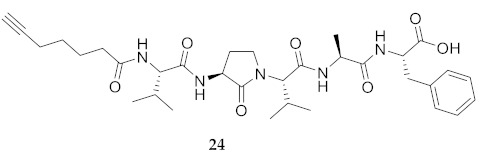



**HCC(CH_2_)_4_CO-Val-Agl-Val-Ala-Phe-OH (24)** Employing the protocol described above for the synthesis of peptide **15** using Fmoc-Agl-Val-OH (462 mg, 3 eq.) in the second coupling, peptide **24** (8.4 mg, 3%) was prepared and shown to be >99% pure LC-MS analysis [50−95% MeOH (0.1% FA) in H_2_O (0.1% FA), 14 min, RT 7.55 min].



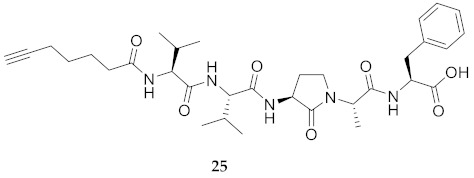



**HCC(CH_2_)_4_CO-Val-Val-Agl-Ala-Phe-OH (25)** Employing the protocol described above for the synthesis of peptide **15** using Fmoc-Agl-Ala-OH (359 mg, 3 eq.) in the first coupling, peptide **25** (9.1 mg, 3%) was prepared and shown to be >99% pure by LC-MS analysis [30−95% MeOH (0.1% FA) in H_2_O (0.1% FA), 14 min, RT 7.94 min].



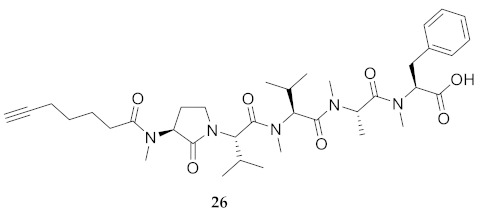



**HCC(CH_2_)_4_CO-Agl(Me)-Val-Val(Me)-Ala-Phe(Me)-OH (26)** Employing the protocol described above for the synthesis of peptide **12** using Fmoc-Agl-Val-OH (462 mg, 3 eq.) in the third coupling, followed by permethylation, peptide **26** (9.5 mg, 3%) was prepared and shown to be >95% pure by LC-MS analysis [50−90% MeOH (0.1% FA) in H_2_O (0.1% FA), 14 min, RT 8.69 min].



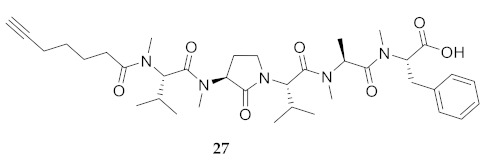



**HCC(CH_2_)_4_CO-Val(Me)-Agl(Me)-Val-Ala(Me)-Phe(Me)-OH (27)** Employing the protocol described above for the synthesis of peptide **12** using Fmoc-Agl-Val-OH (462 mg, 3 eq.) in the second coupling, followed by permethylation, peptide **27** (8.3 mg, 3%) was prepared and shown to be >95% pure by LC-MS analysis [50−90% MeOH (0.1% FA) in H_2_O (0.1% FA), 14 min, RT 9.12 min].



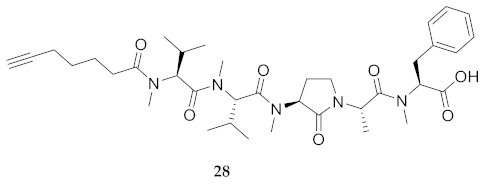



**HCC(CH_2_)_4_CO-Val(Me)-Val(Me)-Agl(Me)-Ala-Phe(Me)-OH (28)** Employing the protocol described above for the synthesis of peptide **12** using Fmoc-Agl-Ala-OH (359 mg, 3 eq.) in the second coupling, followed by permethylation, peptide **28** (6.7 mg, 2%) was prepared and shown to be >95% pure by LC-MS analysis [50−90% MeOH (0.1% FA) in H_2_O (0.1% FA), 14 min, RT 9.01 min].



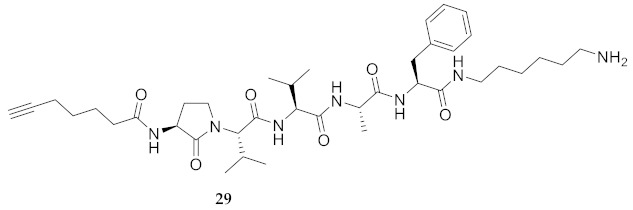



**HCC(CH_2_)_4_CO-Agl-Val-Val-Ala-Phe-NH(CH_2_)_6_NH_2_ (29)** Employing the protocol described above for the synthesis of peptide **18** using Fmoc-Agl-Val-OH (462 mg, 3 eq.) in the fourth coupling, peptide **29** (7.5 mg, 3%) was prepared and shown to be >95% pure by LC-MS analysis [30−95% MeOH (0.1% FA) in H_2_O (0.1% FA), 14 min, RT 6.72 min].



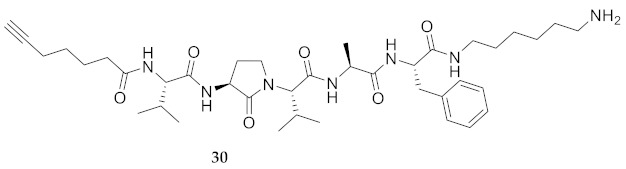



**HCC(CH_2_)_4_CO-Val-Agl-Val-Ala-Phe-NH(CH_2_)_6_NH_2_ (30)** Employing the protocol described above for the synthesis of peptide **18** using Fmoc-Agl-Val-OH (462 mg, 3 eq.) in the third coupling, peptide **30** (8.4 mg, 3%) was prepared and shown to be >95% pure LC-MS analysis [30−95% MeOH (0.1% FA) in H_2_O (0.1% FA), 14 min, RT 6.83 min].



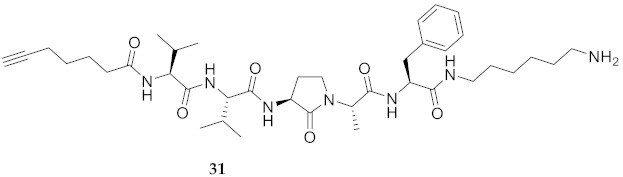



**HCC(CH_2_)_4_CO-Val-Val-Agl-Ala-Phe-NH(CH_2_)_6_NH_2_ (31)** Employing the protocol described above for the synthesis of peptide **18** using Fmoc-Agl-Ala-OH (359 mg, 3 eq.) in the second coupling, peptide **31** (9.1 mg, 3%) was prepared and shown to be >99% pure by LC-MS analysis [30−95% MeOH (0.1% FA) in H_2_O (0.1% FA), 14 min, RT 7.42 min].



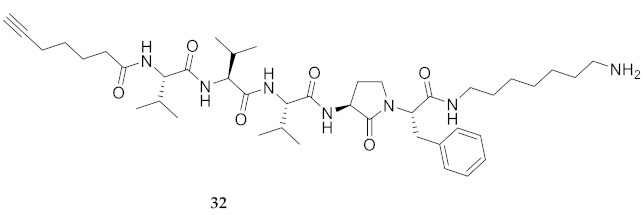





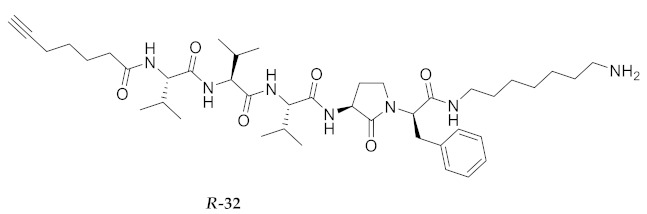



**HCC(CH_2_)_4_CO-Val-Val-Val-Agl-Phe-NH(CH_2_)_6_NH_2_ (32)** and **HCC(CH_2_)_4_CO-Val-Val-Val-Agl-D-Phe-NH(CH_2_)_6_NH_2_ (*R*-32)** Employing the protocol described above for the synthesis of peptide **18** using Fmoc-Agl-(D,L)-Phe-OH (1.03 g, 3 eq.), the mixture of two diastereoisomers was prepared and separated by HPLC on C18 column using a gradient of 50% to 90% MeOH in H_2_O to obtain peptide **32** (1 mg, 1%), which was shown to be >95% pure by LC-MS analysis [50−90% MeOH (0.1% FA) in H_2_O (0.1% FA), 14 min, RT 5.31 min] and peptide **39** (0.9 mg, 1%), which was shown to be >95% pure by LC-MS analysis [50−90% MeOH (0.1% FA) in H_2_O (0.1% FA), 14 min, RT 5.07 min].



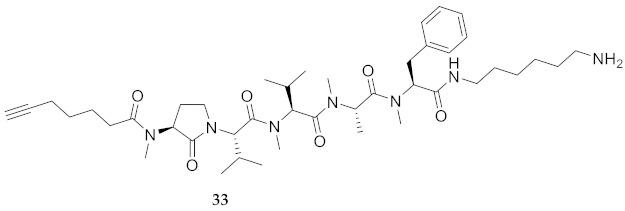



**HCC(CH_2_)_4_CO-Agl(Me)-Val-Val(Me)-Ala(Me)-Phe(Me)-NH(CH_2_)_6_NH_2_ (33)** Employing the protocol described above for the synthesis of peptide **17** using Fmoc-Agl-Val-OH (462 mg, 3 eq.) in the fourth coupling, followed by permethylation and coupling to 1,6-diaminohexane, peptide **33** (9.5 mg, 3%) was prepared and shown to be >99% pure by LC-MS analysis [50−90% MeOH (0.1% FA) in H_2_O (0.1% FA), 14 min, RT 6.72 min].



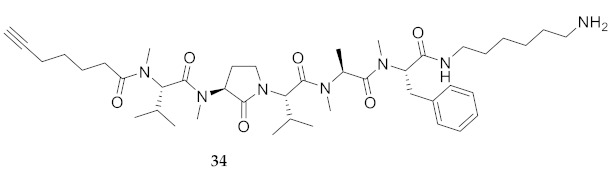



**HCC(CH_2_)_4_CO-Val(Me)-Agl(Me)-Val-Ala(Me)-Phe(Me)-NH(CH_2_)_6_NH_2_ (34)** Employing the protocol described above for the synthesis of peptide **17** using Fmoc-Agl-Val-OH (462 mg, 3 eq.) in the third coupling, followed by permethylation and coupling to 1,6-diaminohexane, peptide **34** (8.3 mg, 3%) was prepared and shown to be >95% pure by LC-MS analysis [50−90% MeOH (0.1% FA) in H_2_O (0.1% FA), 14 min, RT 6.86 min].



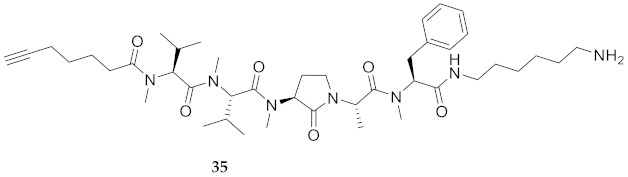



**HCC(CH_2_)_4_CO-Val(Me)-Val(Me)-Agl(Me)-Ala-Phe(Me)-NH(CH_2_)_6_NH_2_ (35)** Employing the protocol described above for the synthesis of peptide **17** using Fmoc-Agl-Ala-OH (359 mg, 3 eq.) in the second coupling, followed by permethylation and coupling to 1,6-diaminohexane, peptide **35** (6.7 mg, 2%) was prepared and shown to be >95% pure by LC-MS analysis [50−90% MeOH (0.1% FA) in H_2_O (0.1% FA), 14 min, RT 6.80 min].



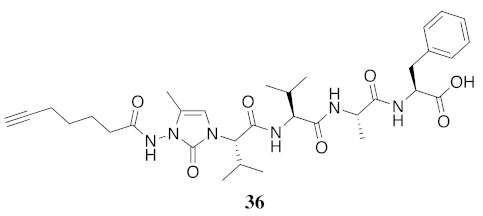



**HCC(CH_2_)_4_CO-(5-Me)Nai-Val-Val-Ala-Phe-OH (36)** Employing the protocol described above for the synthesis of peptide **15** using Fmoc-(5-Me)Nai-Val-OH (318 mg, 2 eq.) in the third coupling, peptide **36** (8.8 mg, 9%) was prepared and shown to be >99% pure by LC-MS analysis [50−90% MeOH (0.1% FA) in H_2_O (0.1% FA), 14 min, RT 5.34 min].



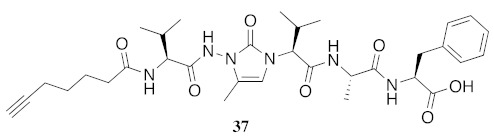



**HCC(CH_2_)_4_CO-Val-(5-Me)Nai-Val-Ala-Phe-OH (37)** Employing the protocol described above for the synthesis of peptide **15** using Fmoc-(5-Me)Nai-Val-OH (318 mg, 2 eq.) in the second coupling, peptide **37** (4.7 mg, 4%) was prepared and shown to be >99% pure by LC-MS analysis [50−90% MeOH (0.1% FA) in H_2_O (0.1% FA), 14 min, RT 4.41 min].

## 5. Conclusions

The unsaturated lipid tail and the *C*-terminal acid and carboxamide functions of almiramide peptides have previously been shown to have relevance for activity against trypanosomatid parasites [24,25]. Moreover, active almiramide analogs possessing sugar amino acids were suggested to adopt bent structures [31]. The influences on anti-parasite activity of amide *N*-methylation and turn-inducing Agl and Nai residues within almiramide peptides have now been investigated in wild type and resistant strains of *L. infatum* and compared with macrophage cytotoxicity. Peptide amides exhibited consistently better activity than their *C*-terminal acid counterparts. Within a set of almiramide 1,6-diaminohexane amides, more potent peptides with relatively high selectivity were typically obtained without methylation (e.g., **16**) and with a single methyl group in MePhe^5^ almiramide **22** than with analogs possessing a methyl amide at other positions and with permethylated analog **17**. On the other hand, permethylated and MeVal^1^ peptide amides **17** and **18** exhibited μM inhibitory activity against the amphotericin B resistant strain with high therapeutic indices. Replacement of amino acid residues by turn-inducing counterparts caused typically losses of activity against the *L. infatum* strains; however, permethylated Agl^2^ amide had similar activity and better selectivity against wild type *L. infatum* compared to permethylated analog **17**. Although studies of the consequences of such structural modifications on metabolism and mechanism of action are in progress, conformers extended about the central residues and mobile at the extremities of the peptide may favor almiramide activity.

## Data Availability

Not applicable.

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
