# Peer review of "Influence of *N*-Methylation and Conformation on Almiramide Anti-Leishmanial Activity"

_molecules, 2021, doi:10.3390/molecules26123606_

Round 1

Reviewer 1 Report

Molecules-1250988.

Influence of N-methylation and conformation on almiramide anti-leishmanial activity

The authors developed an important effort on the synthesis and biological activity of various analogs of almiramides. They succeeded in introducing active analogs of almiramide; obtaining two powerful analogs with high inhibitory activity on a resistant pathogen (Leishmania infantum) obtaining a high therapeutic index. Among other relevant results

Other minor requirements are of importance to support the manuscript (attached file)

Reviewer 2 Report

In this work Nguyen  et al., have prepared and tested a variety of  natural almiramides derivatives as potent antiparasite agents. The work is well designed and executed, although the results are not still so sounds. My suggestion is to be accepted for publication after minor modifications.

Biological activity tests are nor clear how were performed and the way are presented. In table 2, it nor clear if  the results are presented as IC50 or EC50 and how were exactly measured. 

Additionally the evaluation rank (line 209) how was done? was only based on IC50  or in Combination with SI?

Derivatives chemistry should be transferred  to supplementary materials to  make article reading easier. The conclusions which are the most important section of the all manuscript, is coming after a borrowing description of chemical preparation-modifications.

.

Author Response

Biological activity tests are nor clear how were performed and the way are presented. In table 2, it nor clear if the results are presented as IC50 or EC50 and how were exactly measured. 

The results are presented as EC50 values, as stated earlier in the experimental section:

Antileishmanial values in promastigotes were determined by monitoring the growth of parasites after 72 h of incubation at 25 °C in the presence of increasing concentrations of the studied drugs, by measuring A600 using a Cytation 5 multimode reader (BioTek, USA). Drug-efficacy assays were performed with at least three biological replicates from independent cultures (n = 3). EC50 values were calculated based on dose-response curves analyzed by non-linear regression with GraphPad Prism 8.4.3 software (GraphPad Software, La Jolla California, USA).

To further clarify this point, the table title now reads: "Table 2. Structure, bioactivity of almiramide derivatives on different strains of Leishmania (EC50) and cytotoxicity of peptides 12-39." In addition, the caption of Table 2, now reads: “95 % confidence intervals for EC50 determinations are reported within brackets (n = 3).”

Additionally the evaluation rank (line 209) how was done? was only based on IC50  or in Combination with SI?

The evaluation rank was done based on EC50 against wild type L.infatum. To enhance clarity, the text now reads: "Examining peptide potency (EC50)…”. 

Derivatives chemistry should be transferred to supplementary materials to make article reading easier. The conclusions which are the most important section of the all manuscript, is coming after a borrowing description of chemical preparation-modifications.

We have not modified the document as requested, because the manuscript has been prepared for the Special Issue in Honour of Stephen Hanessian and submitted to the section: Organic Chemistry.  In this light, the "description of chemical preparation-modifications” are particularly relevant as recognized by the other Reviewer, who states, “The authors developed an important effort on the synthesis and biological activity of various analogs of almiramides.”  
